# Multi-component reactions via copper(I) difluorocarbene as carbonyl source for constructing α−aminoamide derivatives

Jiuling Li [1,2,6] ✉, Baofan Wang[2,6], Taichen Liu[1,3,6], Qinghong Wen[2], Tongfei Jing[2], Xiang Fu[4], Yingming Pan [3], Kai Wei[1], Xiaoyu Zhou [5] ✉, Wenhao Hu [4] & Zhenghui Kang [2] ✉

Difluorocarbene, an important reactive intermediate in organic synthesis, exhibits intriguing properties and synthetic versatility. However, great challenges in modulating reaction pathways limit its widespread application in synthetic chemistry. While metal-catalyzed difluorocarbene transfer offers a promising strategy but remains a formidable challenge. Herein, we disclose a copper-mediated multicomponent reaction of amine, aldehyde and $BrCF_2CO_2K$ for synthesis of α- aminoamide derivatives, wherein copper-difluorocarbene serve as carbonyl source. Control experiments and DFT calculations support the pathway initiated by formation of a copper-difluorocarbene from $BrCF_2CO_2K$, followed by nucleophilic attack of the amine to produce an ammonium ylide, interception of the ylide with imine, and defluorination via carbonyl migration. This transformation demonstrates broad substrate scope, accommodating not only aromatic aldehydes but also alkyl aldehydes and drug-modified arylamines, highlighting its synthetic applicability. Furthermore, the method provides a practical and ideal alternative to classical Ugi or Strecker reactions, circumventing the need for toxic cyanide salts or unstable isonitriles.

Difluorocarbene represents a versatile synthetic building block, readily accessible from commercially available and inexpensive halodifluoroalkyl reagents[1–11], with wide applications in organic synthesis[1–5], drug development[6–10], and advanced functional materials[11]. As a singlet carbene, difluorocarbene is intrinsically electrophilic due to the existence of an empty *p*-orbital and exhibits conventional carbene reaction properties[1], such as cycloaddition reactions with alkenes or alkynes for formation of *gem*-difluorocyclopropanes[12–17], the Wittig reactions with carbonyl to generate *gem*-difluoroalkenes, etc (Fig. 1a, left)[18–20]. In addition, the difunctional reactions of difluorocarbene[3], in which two chemical bonds are simultaneous formed at the carbene carbon center via reacting with nucleophile, followed by coupling with an electrophile, enable difluorocarbene to be a bipolar $CF_2$ linker and production of the difluoroalkylated compounds (Fig. 1a, left)[21–29]. Compared to these conventional $CF_2$-containing compounds constructing, unconventional transformations of difluorocarbene involving deconstructive functionalization of C-F bonds, beyond its role as a difluoromethyl synthon, have garnered significant attention[4,30]. The reactions are initiated by the electron-deficient characteristics of difluorocarbene that cause the C-F bond scission, enabling difluorocarbene as a versatile C1 synthon for the assembly of valuable N-containing compounds[31–36], heterocycles[31,32], and aliphatic ethers[36] via different

[1]School of Medical Sciences, Pingdingshan University, Pingdingshan, China. [2]Zhongshan Institute for Drug Discovery, Shanghai Institute of Materia Medica, Chinese Academy of Sciences, Zhongshan, China. [3]School of Chemistry and Pharmaceutical Sciences, Guangxi Normal University, Guilin, China. [4]School of Pharmaceutical Sciences, Sun Yat-sen University, Guangzhou, China. [5]School of Pharmaceutical and Chemical Engineering, Taizhou University, Taizhou, China. [6]These authors contributed equally: Jiuling Li, Baofan Wang, Taichen Liu. ✉e-mail: orgchem90@163.com; zhouxiaoyu@tzc.edu.cn; kangzhenghui@simm.ac.cn

**Fig. 1 | Transformations involving difluorocarbene. a** Free difluorocarbene-involved reactions; **b** Pd-catalyzed difluorocarbene transfer via [Pd] = CF$_2$; **c** Cu-catalyzed difluorocarbene transfer via [Cu] = CF$_2$; **d** This work: Cu-catalyzed multi-component reactions via [Cu] = CF$_2$ as carbonyl source.

intermediates such as isocyanides[32], cyano anions[33], formamides[34], and others (Fig. 1a, right)[31,35,36]. However, despite its thought-provoking properties and fascinating application potential, the elusive reactivity of difluorocarbene poses great challenges to the control of the reaction pathways and limits its widespread use in synthetic chemistry, resulting in the restricted reaction types.

In contrast to free difluorocarbenes, metal difluorocarbenes ([M] = CF$_2$) in which transition metals coordinate with carbene carbon to modulate the reactivity offer a promising approach to overcome the limitations mentioned above[37–40]. However, transition-metal catalyzed reactions involving metal–difluorocarbene complexes remain severely underdeveloped, although various [M] = CF$_2$ have been synthesized and characterrized over the past 40 years[37–45]. Recently, Zhang's group successfully synthesized, isolated, and characterized a [Pd$^0$]=CF$_2$ complex for the first time and disclosed its application in the palladium-catalyzed coupling reaction of difluorocarbene with arylboronic acids (Fig. 1b)[46–48]. Following this breakthrough, several catalytic reactions involving palladium-difluorocarbene complexes have been reported, where metal difluorocarbenes exhibit varying reactivity controlled by the valence state of palladium[49–56]. Among them, the strongly nucleophilic [Pd$^0$]=CF$_2$ can be protonated to produce Pd$^{II}$–CF$_2$H, which can subsequently couple with aryl boronic acids[57] and terminal alkynes (Fig. 1b, left)[49,50]. While the electrophilic [Pd$^{II}$]=CF$_2$

undergoes hydrolysis with water to generate CO, serving as a CO surrogate in carbonylation reactions (Fig. 1b, right)[51–57]. Although [Cu]=CF$_2$ complexes were proposed by Burton decades ago and later suggested by Ichikawa in 2016, the development has significantly lagged behind that of [Pd]=CF$_2$ complexes[58–60]. Until 2023, the isolation and structural characterization of [Cu$^I$]=CF$_2$ complex was first achieved by Zhang's group[61].The study demonstrated the electrophilic nature of [Cu$^I$]=CF$_2$ complex, which allows it to be attacked by silyl enol ethers, thereby enabling the catalytic modular synthesis of fluorinated compounds (Fig. 1c)[60–63]. The intriguing discovery of difluorocarbene presents an exciting opportunity to expand the fluorine chemical space. However, catalytic transformations involving [Cu$^I$]=CF$_2$ are still in their infancy, and application of such electrophilic copper difluorocarbene complexes to organic synthesis remains a significant challenge.

Multicomponent reactions (MCRs) are regarded as one of the strategies that most closely approach the 'ideal synthesis', which are flexible, selective, convergent, and atom-efficient processes to construct complex molecules by a single step[64–69]. Our research group has been dedicated to the development of multicomponent reactions involving metal carbene, which proceed through the interception of active ylide/zwitterionic intermediates by various electrophiles, enabling the synthesis of a series of multifunctional molecules[68–75].

However, these reactions predominantly focus on donor-acceptor carbene intermediates. As our continuing interest in multicomponent reactions and considering the intriguing chemical properties of metal difluorocarbene, we propose that metal difluorocarbene could be attacked by nucleophilic reagents to generate a ylide intermediate, which is subsequently captured by an electrophile, thereby enabling a catalytic multicomponent reaction[60,68,69]. This approach is initiated by formation of a difluorocarbene metal complex where metal control the reactivity of difluorocarbene. Thus, it would overcome the limitations of direct difunctionalization of free difluorocarbene that can only couple with limited nucleophiles such as organometallic regents, halide ions and phosphines[3].

Herein, we report a multicomponent reaction of arylamine, aldehyde and $BrCF_2CO_2K$ as difluorocarbene precursor under the catalysis of copper, providing cost-efficient access to multifunctional amide (Fig. 1d). The reaction is proposed to involve copper difluorocarbene intermediate that is attacked by amine to form copper associated ammonium ylide. Subsequent interception of the active ylide intermediate occurs and is accompanied by carbonyl migration with fluorine elimination. In addition, besides aromatic aldehydes, alkyl aldehydes and drug modified arylamine can also be tolerated in this MCRs, which demonstrates the practical applicability of this method. Moreover, this process could serve as an effective and ideal alternative to the Ugi or Strecker reaction, addressing key limitations such as the reliance on highly toxic cyanide salts or the use of toxic and unstable isonitriles[76].

## Results

We initiated our research with the reaction of 4-bromoaniline (**1a**), $BrCF_2COOK$ (**2a**), and benzaldehyde (**3a**) under different reaction conditions (Table 1). The multicomponent defluorination product **4a** was obtained in 57% yield, rather than $CF_2$-containing compounds, when a 2.5:3:1 mixture of **1a**, **2a** and **3a** was treated with CuCl and racemic BINOL-derived phosphoric acid (PPA) in acetonitrile at 50 °C for 12 h under argon atmosphere (Table 1, entry 1). After evaluating a series of copper catalysts (Table S1, in Supplementary Information), $Cu(CH_3CN)_4PF_6$ was selected as the optimal one, leading the **4a** in 78% yield (Table 1, entry 2). Subsequently, various solvents were tested, and it was found that acetonitrile was the ideal solvent for this model reaction (Table S1, in Supplementary Information). When $BrCF_2COOK$ **2a** was replaced by $TMSCF_2Br$ (**2b**) or $BrCF_2COOEt$ (**2c**), no product was detected (Table 1, entries 3, 4). The reaction could proceed smoothly with the lower yield when $ClCF_2COONa$ (**2 d**) was employed as difluorocarbene precursor (Table 1, entry 5). The addition of a Brønsted acid was found to influence the yield of the reaction (Table 1, entries 6–11, and Table S2, in Supplementary Information). Among the acids tested, p-toluenesulfonic acid (TsOH) proved to be the most effective, affording **4a** with in 83% yield. (Table 1, entry 11). The optimal reaction temperature was determined to be 50 °C. When the temperature was lowered, even with extended reaction times, the starting materials remained unchanged (Table 1, entries 12, 13). Additionally, increasing the temperature did not lead to a significant increase in the

## Table 1 | Condition optimization[a]

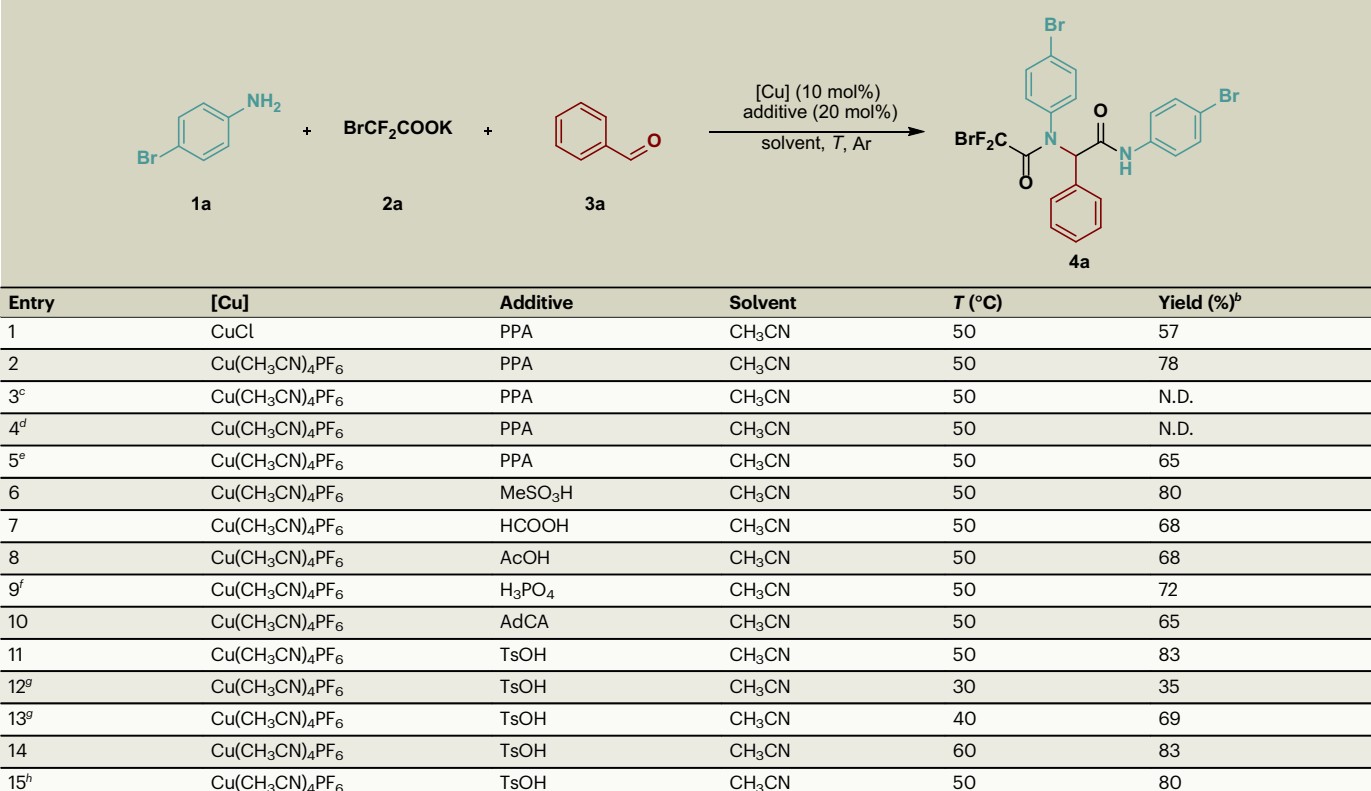

| Entry | [Cu] | Additive | Solvent | T (°C) | Yield (%)[b] |
|---|---|---|---|---|---|
| 1 | CuCl | PPA | $CH_3CN$ | 50 | 57 |
| 2 | $Cu(CH_3CN)_4PF_6$ | PPA | $CH_3CN$ | 50 | 78 |
| 3[c] | $Cu(CH_3CN)_4PF_6$ | PPA | $CH_3CN$ | 50 | N.D. |
| 4[d] | $Cu(CH_3CN)_4PF_6$ | PPA | $CH_3CN$ | 50 | N.D. |
| 5[e] | $Cu(CH_3CN)_4PF_6$ | PPA | $CH_3CN$ | 50 | 65 |
| 6 | $Cu(CH_3CN)_4PF_6$ | $MeSO_3H$ | $CH_3CN$ | 50 | 80 |
| 7 | $Cu(CH_3CN)_4PF_6$ | HCOOH | $CH_3CN$ | 50 | 68 |
| 8 | $Cu(CH_3CN)_4PF_6$ | AcOH | $CH_3CN$ | 50 | 68 |
| 9[f] | $Cu(CH_3CN)_4PF_6$ | $H_3PO_4$ | $CH_3CN$ | 50 | 72 |
| 10 | $Cu(CH_3CN)_4PF_6$ | AdCA | $CH_3CN$ | 50 | 65 |
| 11 | $Cu(CH_3CN)_4PF_6$ | TsOH | $CH_3CN$ | 50 | 83 |
| 12[g] | $Cu(CH_3CN)_4PF_6$ | TsOH | $CH_3CN$ | 30 | 35 |
| 13[g] | $Cu(CH_3CN)_4PF_6$ | TsOH | $CH_3CN$ | 40 | 69 |
| 14 | $Cu(CH_3CN)_4PF_6$ | TsOH | $CH_3CN$ | 60 | 83 |
| 15[h] | $Cu(CH_3CN)_4PF_6$ | TsOH | $CH_3CN$ | 50 | 80 |

*N.D.* not detected, *AdCA* 1-adamantanic acid.

[a]Unless otherwise noted, all reactions were conducted with 0.2 mmol of **3a** in acetonitrile (3 mL) for 12 h, **1a**: **2a**: **3a** = 2.5: 3: 1 at the corresponding temperature under argon atmosphere.

[b]Isolated yield.

[c]**2a** was replaced by $TMSCF_2Br$ (**2b**).

[d]**2a** was replaced by $BrCF_2COOEt$ (**2c**).

[e]**2a** was replaced by $ClCF_2COONa$ (**2d**).

[f]Purity of $H_3PO_4$: 85 wt. % in $H_2O$.

[g]Reaction time: 16 h.

[h]The reaction occurred in air atmosphere.

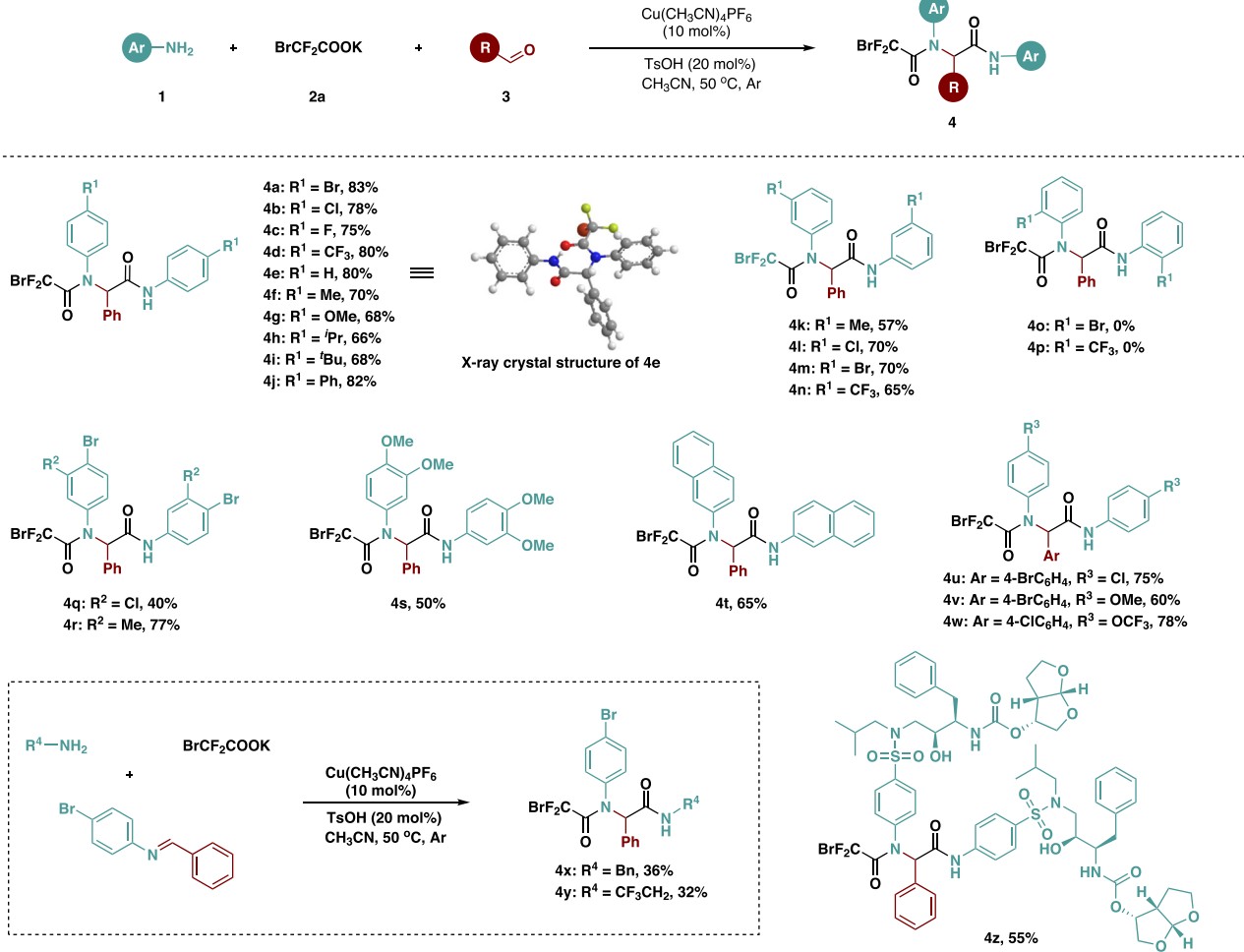

**Fig. 2 | Substrate scope of amines.** Unless otherwise noted, all reactions were conducted with 0.2 mmol of **3** in acetonitrile (3 mL) at 50 °C for 12 h, **1**: **2a**: **3** = 2.5: 3: 1 under argon atmosphere. Isolated yield.

yield of the reaction (Table 1, entry 14). When the reaction occurred in air atmosphere, the yield of multicomponent product **4a** would slightly decrease (Table 1, entry 15). Attempts at the asymmetric synthesis of the target product **4a** were systematically explored. However, no significant stereoselectivity was observed in the transformation (Table S3, S4, in Supplementary Information).

With the optimized reaction conditions in hand, we explored the scope of the multi-component reaction by investigating aromatic amines **2** (Fig. 2). The reaction with substrates containing *para*-substituents on the aromatic ring of aniline proceeded smoothly, and whether halogen (**4a-4c**) and trifluoromethyl (**4 d**) substituents with electron withdrawing properties or alkyl (**4 f**, **4 h** and **4i**), methoxy (**4 g**), and phenyl (**4 j**) substrates with electron withdrawing properties gave the corresponding products in middle to good yields. The structure of **4e**, derived from benzaldehyde, aniline, and BrCF₂COOK, was unambiguously confirmed by X-ray crystallographic analysis. In addition, *meta*-substituents on the aromatic ring of aniline were well tolerated in such transformation, affording the multi-component products (**4k-4n**). It is worth noting that the *ortho*-substituted products (**4o** and **4p**) were not detected, likely due to steric hindrance from the functional groups. Both disubstituted aromatic amines and β-naphthylamine were suitable for this reaction system (**4q-4t**). When halogen substituted aromatic aldehydes are used as substrates, aniline substituted with chlorine, methoxy, and trifluoromethoxy are also suitable substrates, yielding the corresponding products in good yields (**4u-4w**). Furthermore, aliphatic amines such as benzylamine

(BnNH₂) and 2,2,2-trifluoroethylamine (CF₃CH₂NH₂) were also evaluated as substrates in our reactions, and the reactions underwent well to react with BrCF₂CO₂K and imine as the reaction partner. The desired products were obtained in 36% and 32% yield, respectively (**4x** and **4 y**), albeit with competitive formation of aniline-derived products **4a**. Excitingly, drug Prezista as HIV-1 protease inhibitor could also be applied to this transformation, and the target compound was obtained in 55% yield (**4z**).

The scope of aldehydes was then investigated (Fig. 3). When aniline was used as the substrate, both monosubstituted 4-trifluoromethylbenzaldehyde and disubstituted 3,4-dichlorobenzaldehyde were well tolerated in this reaction system, to afford the target compounds (**4aa** and **4ab**). When 4-bromoaniline participated in the reaction, aromatic aldehyde derivatives with electron-donating or electron-withdrawing functional groups on the *para*-position of aromatic ring, including halo, trifluoromethyl, nitro, methyl, methoxy, phenyl, and thiomethyl were harmonious with such catalytic system, yielding corresponding products (**4ac-4aj**). Once *ortho*- and *meta*-substituted aromatic aldehydes were applied to the reaction, the yields of the corresponding products (**4ak-4aq**) were favorable without significant decrease, which indicated that the steric hindrance of aromatic aldehydes did not interfere with the reaction yield. Among them, the structure of **4ap** was unambiguously confirmed by X-ray crystallographic analysis. Heterocyclic aromatic aldehydes, polycyclic aromatic aldehyde and fused heterocyclic aromatic aldehydes were well tolerated in such reaction system, giving the

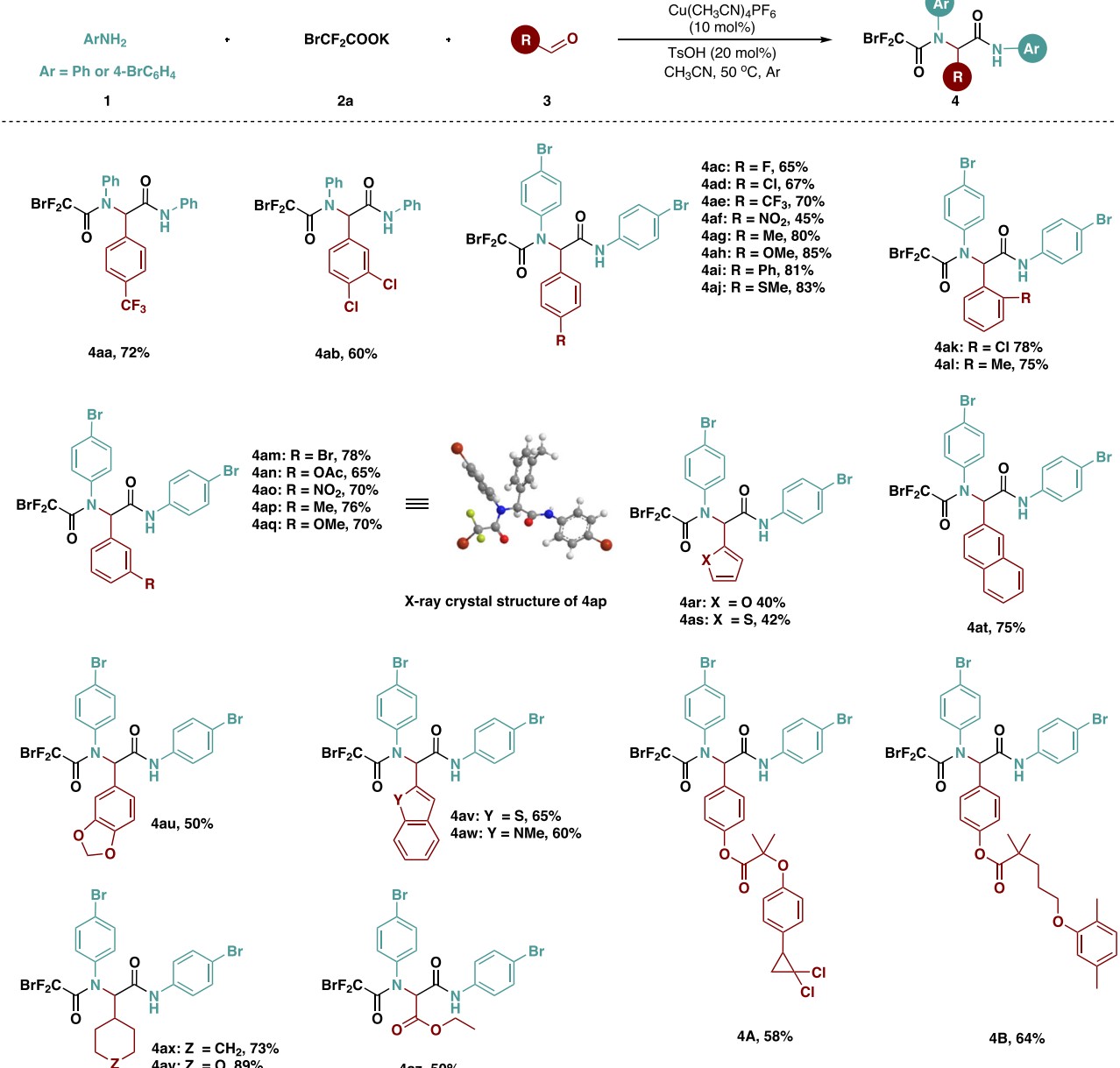

**Fig. 3 | Substrate scope of aldehydes.** Unless otherwise noted, all reactions were conducted with 0.2 mmol of **3** in acetonitrile (3 mL) at 50 °C for 12 h, **1**: **2a**: **3** = 2.5: 3: 1 under argon atmosphere. Isolated yield.

multi-component products (**4ar-4aw**) in middle to good yields. To our delight, non-aromatic aldehydes such as cyclohexanecarbaldehyde, tetrahydropyran-4-carbaldehyde, and glyoxylic esters are also suitable substrates, affording desired products (**4ax-4az**). Moreover, aldehydes **3ab** and **3ac** derived from Gemfibrozil and Ciprofibrate were also suitable for this reaction system, to give the desired products (**4 A** and **4B**), which demonstrated the utility of this reaction.

Moreover, to highlight the synthetic value of such approach, when 4-methoxybenzaldehyde was scaled up to 6 mmol, the desired amide **4ah** was smoothly obtained in 67% yield, and the amides could be easily further functionalized (Fig. 4). First, compound **4ah** was converted into reduced product **5** with 92% yield under the action of NaBH₄. Then, compound **4ah** could smoothly generate cyclization products with 90% yield in the presence of DBU. In addition, when amide **4 g** was used as starting material, the reaction yielded the product **7** with 55% yield, which was deprotected from the *p*-methoxyphenyl group (PMP) with the assistance of ceric ammonium nitrate

(CAN). Of note, when aldehydes were replaced by *N*, *N*-dibenzyl-1-methoxymethanamine **8**, corresponding multi-component products **9** was obtained with 58% isolated yield.

In order to better understanding the pathway for this transformation, several validation experiments were conducted. Firstly, 100 μL oxygen-18 water was added to template reaction, the yield of multi-component products was not affected, and no ¹⁸O labeled products were observed by HRMS. The standard template reaction results in roughly the same yield with or without water (100 μL). These results indicated that water is not involved in the reaction and the water molecules generated in situ were not the oxygen source of the product in multi-component reactions (Fig. 5a). Secondly, when benzaldehyde was omitted from the multi-component reactions, *p*-tert-butylaniline **1i** was reacted with BrCF₂COOK under the standard conditions, leading to formylation products **10a** formed via N-H insertion followed by defluorination and direct amidation product **11a** with nearly 1:1 ratio in yield (Fig. 5b). Notably, isonitrile was not detected by LC-MS at any

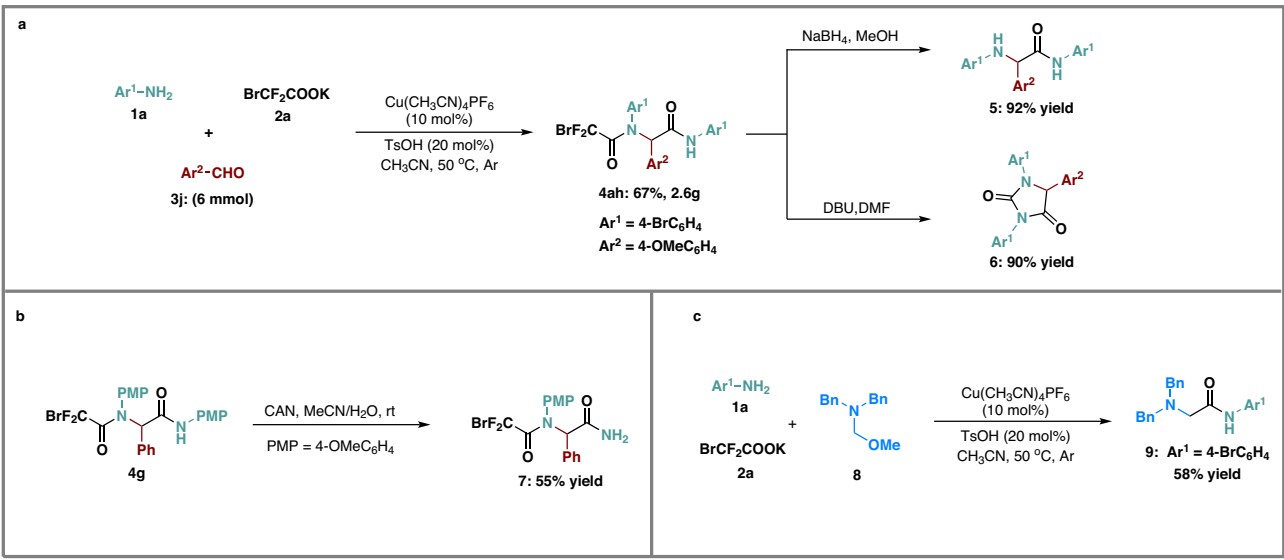

**Fig. 4 | Synthetic application. a** Gram-scale reactions and transformations of the product **4ah**; **b** Remove the *p*-methoxyphenyl group in **4 g**; **c** Multi-component reaction with dibenzyl-1-methoxymethanamine. DBU = 1,8-Diazabicyclo[5.4.0]undec-7-ene, CAN ceric ammonium nitrate.

a) Control reaction with/ without water and ¹⁸O-isotope labeling

b) Control reaction of aniline with BrCF₂COOK

c) Cross-multicomponent reaction with **10a** and **11a**

d) Control reaction without copper

e) Competition experiment of BrCF₂CO₂K, 4-bromoaniline **1a** and silyl enol ether **13a**

**Fig. 5 | Control experiments. a** Control reaction with/ without water and ¹⁸O-isotope labeling; **b** Control reaction of aniline with BrCF₂CO₂K; **c** Cross-multicomponent reaction with **10a** and **11a**; **d** Control reaction without copper; **e** Competition experiment of BrCF₂COOK, 4-bromoaniline **1a** and silyl enol ether **13a**.

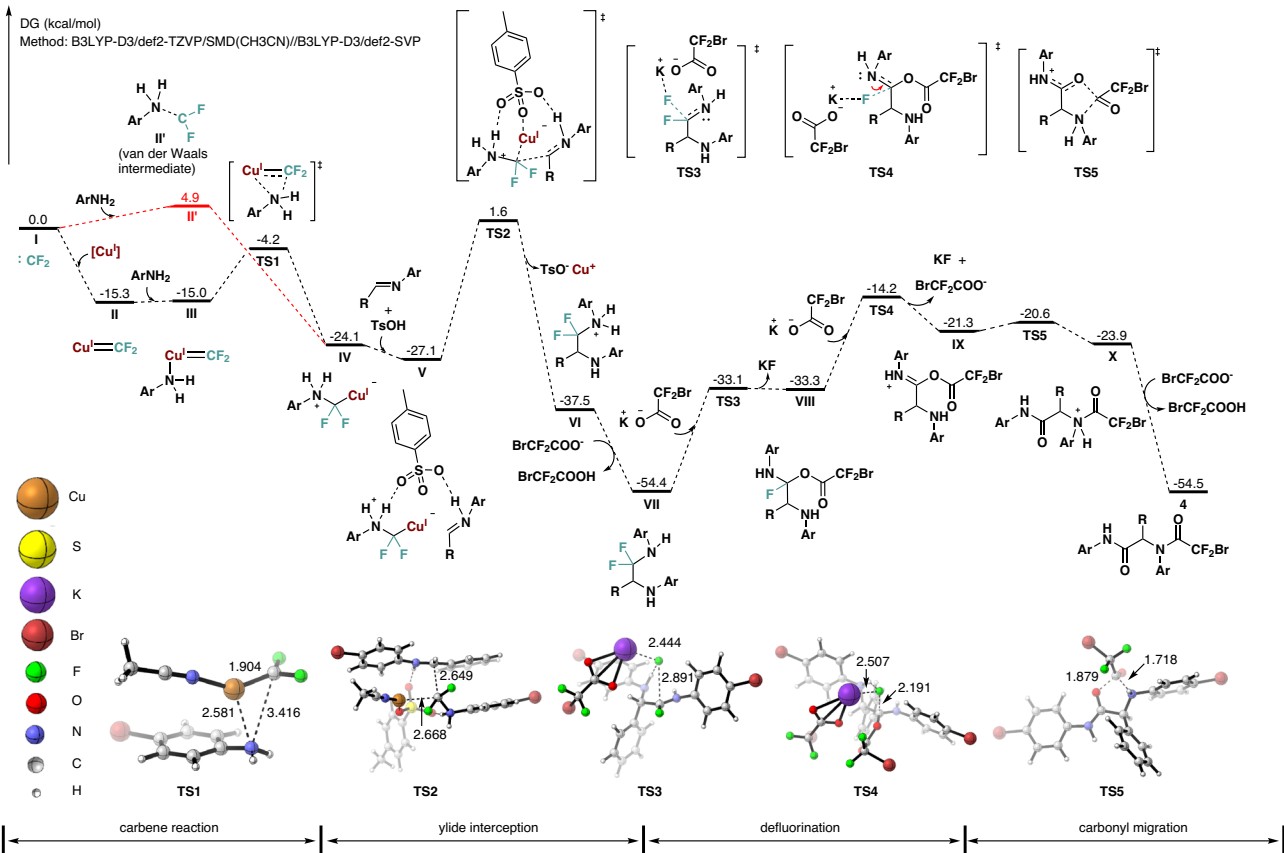

**Fig. 6 | Proposed reaction mechanism and the DFT calculated free-energy profile from difluorocarbene to the product.** Copper-difluorocarbene was demonstrated as the key intermediate rather than free difluorocarbene for formation of ylide intermediate **IV**, and subsequently ylide interception, defluorination and carbonyl migration process were energetic feasibility.

stage of the reaction. According to the literature research, isonitriles can only be obtained in the presence of organic or inorganic bases. However, our optimal reaction conditions were not conducive to the generation of isonitriles[30]. Therefore, we speculated that there was no involvement of isocyanide intermediates in such transformation. Then, compounds **10a** and **11a** were added to the model reaction of 4-bromoaniline **1a**, BrCF$_2$COOK **2a**, and benzaldehyde **3a**, only the target product **4a** was obtained, with no detectable formation of cross-multicomponent products bearing 4-tert-butyl substitution (Fig. 5c). This observation suggests that neither the formylation product **10a** nor the amidation product **11a** serve as intermediates in this trans-formation, and exclude a stepwise mechanism for this multi-component reaction.

Next, the role of the copper in the catalytic process was further investigated. The control experiments of 4-bromoaniline **1a**, BrCF$_2$COOK **2a**, and benzaldehyde **3a** were performed under the standard conditions in the absence of copper or with alternative Lewis acidic metal catalysts, including Sc(OTf)$_3$, Yb(OTf)$_3$, Zn(OTf)$_2$, Ni(OTf)$_2$ and AgSbF$_6$. Notably, no significant formation of the target product was observed in any of these cases, even at elevated temperatures (Table S5, in Supplementary Information). These findings demonstrate that the copper catalyst fulfills a critical and distinctive role beyond conventional Lewis acid catalysis. Previous studies have unequivocally demonstrated that free difluorocarbenes readily undergo cycloaddition with silyl enol ethers to form gem-difluorocyclopropanes, whereas copper difluorocarbene species exhibit complete incapacitation of cyclopropanation activity[60,61]. Capitalizing on this distinct reactivity profile, the competitive experiments of BrCF$_2$COOK, 4-bromoaniline and silyl enol ether **13a** with or without copper catalyst were carried out. The formylation product **10b** was obtained in 37% yield with no

detectable cyclopropanation product **12a** under the catalysis of Cu(CH$_3$CN)$_4$PF$_6$. In contrast, in the absence of copper catalyst, [19]FNMR analysis revealed the formation of difluorocyclopropane **12a** in 19% yield, alongside a significant reduction in formylation product yields (**10b** < 5%, Fig. 5e). In addition, the template reaction of BrCF$_2$COOK, 4-bromoaniline and benzaldehyde was also conducted under the previously reported conditions for transformations involving a copper-difluorocarbene intermediate (copper salt, 2,9-diMe-1,10-phen, CH$_3$CN, 50 °C.)[60,62], yielding the desired multi-component product in 60% yield (Figure S5, in Supplementary Information). These results are consistent with prior reports, corroborating the generation of a copper-difluorocarbene intermediate in our reaction system, and supporting the potential involvement of a copper-difluorocarbene species in the present multi-component reaction. We performed density functional theory (DFT) calculations to illustrate the formation of the intermediate **IV** as shown in Fig. 6. The free difluorocarbene **I** preferentially coordinate with [Cu$^I$]$^+$ to form the copper-difluorocarbene complex **II**, which is energetically favorable (−15.3 kcal/mol) compared to the direct interaction with ArNH$_2$, yielding a weakly bound van der Waals complex **II'** (4.9 kcal/mol). Subsequent coordination of ArNH$_2$ to copper-difluorocarbene complex **II** generates intermediate **III** (−15.0 kcal/mol), which undergoes C-N bond formation via transition state **TS1** (−4.2 kcal/mol) to afford the copper-associated ammonium ylide intermediate **IV** (−24.1 kcal/mol). The overall process for formation the intermediate **IV** via copper difluorocarbene isexergonic, releasing 24.1 kcal/mol, indicating a thermodynamically favorable pathway. In the absence of copper catalyst, formation of a weakly bound van der Waals intermediate **II'** from free difluorocarbene with ArNH$_2$ exhibits significantly slower kinetics (4.9 kcal/mol versus −15.3 kcal/mol for the copper-mediated pathway).

Computational attempts to locate a transition state for formation of C-N bond from **II′** were unsuccessful, excluding the possibility of intermediate **IV** forming directly from free difluorocarbene (Figs. S7, S8, in Supplementary Information). The DFT calculations highlight the crucial role of copper(I) in the transformation, which initiates with coordination of copper(I) to difluorocarbene, generating a key copper-difluorocarbene intermediate. The key intermediate subsequently undergoes C-N bond formation through nucleophilic attack by ArNH₂, ultimately affording intermediate **IV**.

On the basis of the above results and previous works[30,68,69], the plausible subsequently reaction pathway involving ylide interception, defluorination and carbonyl migration of the multicomponent reaction is proposed and demonstrated its energetic feasibility through DFT calculations (Fig. 6). The nucleophilic ammonium ylide intermediate **IV** is captured by the activated imine via transient state **TS2** (1.6 kcal/mol), in which p-toluenesulfonic acid and bromodifluoroacetate ion facilitate the proton transfer, leading to the product **VII** (−54.4 kcal/mol) with releasing the copper(I). The resulting product **VII** containing a fragment of CF₂ adjacent to nitrogen atom is vulnerable, and the consecutive scission of Csp³-F bond occurs under the assistant of BrCF₂COOK. According to the DFT, the monofluoroimine species **VIII** (−33.3 kcal/mol) is formed via nucleophilic substitution of bromodifluoroacetate ion with **VII** through transition state **TS3** (−33.1 kcal/mol), followed by the second Csp³-F bond cleavage to generate intermediate **IX** (−21.3 kcal/mol). Intramolecular carbonyl migration of intermediate **IX** via **TS5** (−20.6 kcal/mol) with a barrier of 0.7 kcal/mol and subsequent deprotonation eventually render α-aminoamide products.

## Discussion

In summary, we have developed copper-catalyzed MCRs for synthesis of multifunctional amide derivatives from amine, aldehyde and BrCF₂COOK without the need for any ligands. The mild reaction conditions, non-toxic nature, and use of readily available raw materials demonstrate that this reaction serves as an effective alternative strategy to the Strecker or Ugi reactions, enabling the synthesis of versatile and valuable products. Additionally, the high functional group tolerance, accommodating not only aromatic aldehydes but also alkyl aldehydes and even complex drug-like molecules, underscores the practical applicability of this method. Control experiments and DFT calculations systematically support that the copper difluorocarbene complex serves as the key intermediate in this transformation and acts as the carbonyl source for the formation of the amide group, and exclude the formation of isonitriles under the reaction conditions. The reaction is proposed to through the formation of copper difluorocarbene, nucleophilic attack by the amine to produce a copper-associated ammonium ylide, interception of the active ylide intermediate with imine, and subsequent defluorination via carbonyl migration. This sequence accounts for the overall high efficiency and distinctiveness of the reaction.

## Methods

### General

All ¹H NMR (500 MHz, 600 MHz) and ¹³C NMR (125 MHz, 150 MHz) and ¹⁹F NMR (471 MHz) spectra were recorded on 500 or 600 MHz spectrometers in in CDCl₃, DMSO-$d_6$ and Methanol-$d_4$. Chemical shifts were reported in ppm with the solvent signal as reference, and coupling constants (*J*) were given in Hertz. The peak information was described as: s = singlet, d = doublet, t = triplet, q = quartet, m = multiplet, br = broad. High-resolution mass spectrometry (HRMS) was recorded on a commercial apparatus (ESI Source). Single crystal X-ray diffraction data were recorded on Bruker-AXS SMART APEX II single crystal X-ray diffractometer.

### General procedure for synthesis of product 4

To an oven-dried 10 mL Schlenk tube equipped with a stir bar was added Cu(CH₃CN)₄PF₆ (7.5 mg, 0.02 mmol, 10.0 mol%), TsOH (6.9 mg, 0.04 mmol, 20.0 mol%), aromatic amines **1** (0.5 mmol, 2.5 equiv), BrCF₂COOK **2a** (128 mg, 0.6 mmol, 3.0 equiv), and aldehydes **3** (0.2 mmol, 1.0 equiv), and suspended in CH₃CN (3.0 mL) under dry argon atmosphere. The resulting mixture was stirred at 50 °C for 12 hours. The progress of the reaction was monitored by TLC. After the reaction was complete, the reaction was cooled to room temperature and concentrated under reduced pressure. The residue was purified by flash column chromatography (eluent: EA:PE = 1/20 ‑ 1/5) to give the pure product **4**.

### General procedure for synthesis of product 9

To an oven-dried 10 mL Schlenk tube equipped with a stir bar was added Cu(CH₃CN)₄PF₆ (7.5 mg, 0.02 mmol, 10.0 mol%), TsOH (6.9 mg, 0.04 mmol, 20.0 mol%), aromatic amine **1a** (0.3 mmol, 1.5 equiv), BrCF₂COOK **2a** (128 mg, 0.6 mmol, 3.0 equiv), and *N,N*-dibenzyl-1-methoxymethanamine **8** (0.2 mmol, 1.0 equiv), and suspended in CH₃CN (2.0 mL) under dry argon atmosphere. The resulting mixture was stirred at 50 °C for 12 hours. The progress of the reaction was monitored by TLC. After the reaction was complete, the reaction was cooled to room temperature and concentrated under reduced pressure. The residue was purified by flash column chromatography (eluent: EA:PE = 1/20 ‑ 1/10) to give the pure product **9** (58% yield).

## Data availability

All data supporting the findings described in this manuscript are available in the the main text and Supplementary Information. For full characterization data of new compounds and experimental details, see Supplementary Methods. ¹H NMR, ¹³C NMR, and ¹⁹F NMR spectra are supplied for all new compounds. The cartesian coordinates of the optimized structures in this study are provided in Source Data file. Source data are provided with this paper. Crystallographic data for the structures reported in this Article have been deposited at the Cambridge Crystallographic Data Center, under deposition numbers CCDC2393043 (**4e**) and CCDC2393055 (**4ap**). Copies of the data can be obtained free of charge via https://www.ccdc.cam.ac.uk/structures/. All other data are available from the corresponding authors upon request. Source data are provided with this paper.

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

## Acknowledgements

Financial support from Natural Science Foundation of Guangdong Province (2023A1515110403 T.J., 2024A1515030037 Z.K.) is gratefully acknowledged. We also acknowledge financial support from the China Postdoctoral Science Foundation under Grant (NO. 2023M743925 T.J.), Natural Science Foundation of Henan Province (242300420201 K.W., 252300420796 J.L.) and Natural Science Foundation of Zhejiang Province (LQN25B020014 X.Z.) for their financial support.

## Author contributions

J.L., B.W. and T.L. contributed equally to this work. Z.K. and J.L. contributed to the conception and design of the experiments. Z.K. directed the project. B.W., T.L., Q.W. and T.J. performed the experiments and analyzed the data. X.Z and X.F. contributed to density functional theory (DFT) calculations. Z.K., J.L. and W.H. wrote the manuscript. Y.P. and K.W. provided valuable suggestions to the project. All authors discussed the results and commented on the manuscript.

## Competing interests

The authors declare no competing interests.
