## [Transparent Peer Review file · Nature Communications]

Multi-Component Reactions via Copper(I) Difluorocarbene as Carbonyl Source for Constructing α - Aminoamide Derivatives

Corresponding Author: Dr Zhenghui Kang

Version 0:

Reviewer comments:

Reviewer #1

(Remarks to the Author)

As a versatile synthon, difluorocarbene has received increasing attention recently. While metal difluorocarbene-involved catalytic coupling remains challenging due to the lack of understanding of metal difluorocarbene chemistry. The manuscript by Kang and coworkers reports a copper-catalyzed multi-component reaction via a difluorocarbene pathway. This is a good extension of the pioneering work reported by Zhang (ref 65). The approach uses $\text{BrCF}_2\text{CO}_2\text{K}$ as the difluorocarbene precursor, arylamines as the nucleophiles, and aldehydes as the electrophiles, providing a series of α -aminoamide derivatives with moderate to good yields. However, the reaction mechanism is unclear. The authors conducted several experiments to conclude that a copper difluorocarbene species is involved in the reaction. However, the sole control reaction without copper salt cannot support a pathway involving copper difluorocarbene (Fig. 5b). As the difluorocarbene is electrophilic, the direct reaction of arylamine with difluorocarbene, followed by a combination with copper salt, can also generate intermediate I. More evidence to support the copper difluorocarbene-involved pathway would make readers understand the chemistry better. The formation of compound X demonstrates that the attack of arylamine on the difluorocarbene is feasible. The resulting adduct may react with copper salt to generate I. Additionally, it has been shown that copper(I) difluorocarbene is very sensitive to H_2O and decomposes immediately (ref 65). However, the current approach is not sensitive to H_2O , indicating that a copper difluorocarbene might not be involved in the reaction. Overall, this is an interesting reaction. Using difluorocarbene, arylamines, and aldehydes provides an efficient route to α -aminoamide derivatives. The manuscript can be published in Nat. Commun., but revisions are needed.

1) More mechanistic studies are needed to shed light on the reaction mechanism.

2) How about aliphatic amines? The applicability of such amines would improve the synthetic utility of the approach.

3) The sentence presented in the abstract needs to be revised.

“However, its high reactivity and the uncontrollable reaction pathways it triggers limit its widespread application in synthetic chemistry”

Page 2, line 40, “fellow” should be “followed”

Reviewer #2

(Remarks to the Author)

This manuscript describes a copper-catalyzed synthesis of α -aminated amides by connecting arylamines and aromatic or alkyl aldehydes with a carbonyl moiety (CO) derived from a copper difluorocarbene complex. Although the key intermediate of this transformation, a copper difluorocarbene complex, generated from commercially available $\text{BrCF}_2\text{CO}_2\text{K}$ and a copper(I) salt, has already been reported, the authors demonstrated a new way to its application. Furthermore, the reaction proceeds with a broad substrate scope and high functional group tolerance. Because interesting and valuable results are included in this manuscript, its publication in NCOMMS is recommended. Before its acceptance, the authors may consult the following:

(1) Pages 2-3, lines 64-65: The method for the generation of a copper difluorocarbene complex from $\text{BrCF}_2\text{CO}_2\text{K}$ and a copper(I) salt was initially reported in Ref. 61, which Zhang followed in Refs. 65-66. These facts should be clearly described

in the manuscript.

(2) Page 3, lines 66-68: In the reactions reported in Refs. 62-64, $\text{BrCF}_2\text{CO}_2\text{K}$ and a copper(I) salt are used. However, copper difluorocarbene complexes are not involved, but $\text{CF}_2\text{CO}_2\text{R}$ radical.

(3) Page 3, lines 81-83: The concept: copper difluorocarbene complexes are electrophilic to be attacked by nucleophiles to generate anionic intermediates, which in turn react with electrophiles, is reported in Ref. 61 as a method for ring construction. This fact should be shown in the manuscript by quotation.

(4) Page 5, line 124: "4a" should be typed boldly.

(5) Page 8, line 158: For "cyclohexanol," read "cyclohexanecarbaldehyde."

(6) Page 8, line 158: For "morphological," read "tetrahydropyran-4-carbaldehyde."

(7) Page 8, line 158: For "aldehyde esters," read "glyoxylates" or "glyoxylic esters."

(8) Page 9, line 186: Ref. 61 should be quoted before Refs. 65-66 for the method of copper difluorocarbene generation.

(9) Page 9, line 189; Page 10, Fig. 5c: How about alkylamines (RNH_2)? Copper difluorocarbene complexes seem to react with alkylamines to afford isonitriles. If not, alkylamines could be added in Fig. 2 to broaden the substrate scope.

Version 1:

Reviewer comments:

Reviewer #1

(Remarks to the Author)

The authors have effectively addressed the concerns raised by the reviewers. However, the discussion of copper difluorocarbene chemistry in the introduction requires revision. Copper difluorocarbene species was initially proposed by Burton (JACS 1986, 832; and Ref 18). This complex was also suggested by Ichikawa (Ref 61), but there is no direct evidence to confirm the involvement of a $[\text{Cu}(\text{I})]=\text{CF}_2$ species in the reaction. It would be more appropriate to integrate the description of Ichikawa's work into the sentence: "Although $[\text{Cu}]=\text{CF}_2$ complexes were proposed over a decade ago, their development has significantly lagged behind that of $[\text{Pd}]=\text{CF}_2$ complexes."

Reviewer #2

(Remarks to the Author)

This manuscript describes a copper-catalyzed synthesis of α -aminated amides by connecting arylamines and aromatic or alkyl aldehydes with a carbonyl moiety (CO) derived from a copper difluorocarbene complex. Because the manuscript was adequately revised, its publication in NCOMMS is recommended. Before its acceptance, the authors may consult the following:

bromodifluoroacetic anion

p-tert butylaniline 1i

(1) Pages 3, line 7: Quotation "62-64" should be "61-64."

(2) Page 4, Fig 1d: The imine should be the corresponding iminium above the arrow.

(3) Page 5, line 5 from the bottom: For "underwent well with to react," read "underwent well to react."

(4) Page 9, line 1: For "p-methylphenyl group (PMP)" read "p-methoxyphenyl group (PMP)."

(5) Page 9, line 9 from the bottom: For "p-tert butylaniline 1i" read "4-tert-butylaniline 1i."

(6) Page 12, line 13: For "bromodifluoroacetic anion" read "bromodifluoroacetate ion."

(7) Page 12, line 13: For "and follow by" read "followed by."

(8) Page 12, lines 14-16: $\text{S}_{\text{N}}\text{Ar}$ substitution with Csp³-F bond should be explained more clearly or by adding a scheme in Fig. 6.

Reviewer #3

(Remarks to the Author)

In the manuscript, the authors contributed a copper-catalyzed multi-component reaction that efficiently constructs a series of α -aminoamide derivatives using difluorocarbene as a carbonyl source. The authors have preliminarily explored the reaction mechanism through experiments and DFT calculations, proposing that the reaction proceeds via a copper(I) difluorocarbene intermediate. Following the first review, the authors supplemented their work with additional control experiments and DFT calculations to further support the proposed mechanism. The computational part provides some theoretical support for understanding the reaction pathway, particularly for the formation of the copper(I) difluorocarbene intermediate and the subsequent nucleophilic attack by the amine. The manuscript is recommended for publication in Nat. Commun. after further improvements addressing several issues related to theoretical calculations.

1. The authors specified in the Supporting Information that the DFT calculations were carried out using the M06-L functional, with geometry optimizations performed with the def2-SVP basis set and single-point energy calculations conducted with the def2-TZVP basis set under the SMD solvation model (acetonitrile). However, for copper-catalyzed reactions, particularly those involving complex electronic structures, it is recommended to compare the results obtained with other functionals—such as $\omega\text{B97X-D}$, B3LYP-D3(BJ), or double-hybrid functionals—in order to assess the influence of functional choice on the reliability of the computational results.

2. In the complete catalytic cycle proposed by the authors (Fig. 6), intermediate IV subsequently reacts with an imine to form V, followed by crucial defluorination and carbonyl migration steps to generate VI and the final product. The current computational study does not cover these critical transformation steps. Particularly, the defluorination step (C-F bond

cleavage) and carbonyl migration are vital for a complete understanding of the reaction mechanism. The lack of computational data for these steps leaves the latter half of the mechanism largely speculative. It is recommended that the authors supplement their work with DFT calculations for these key elementary steps, including locating the corresponding transition states and intermediates, and providing the energy profiles.

3. The first reviewer mentioned that copper(I) difluorocarbene is sensitive to water. The authors explained in their rebuttal that the nucleophilicity of the amine is superior to that of water, and the excess carbene precursor and amine allow the reaction to tolerate trace amounts of water. Although experimental controls (e.g., using ^{18}O -labeled water) ruled out water as the oxygen source for the product, from a computational perspective, comparing the energy barriers for the attack of amine versus water on the copper(I) difluorocarbene could quantitatively assess the possibility and rate differences of these competitive reactions, thereby providing stronger theoretical support for the experimental observations.

4. When discussing energies, it is recommended to clearly distinguish between Electronic Energy, Gibbs Free Energy, and Enthalpy. The Supporting Information mentions that "refining of single point energies with solvent effect was calculated at the M06L and def2-TZVP level of theory," which usually refers to electronic energies. Have the Gibbs free energy corrections obtained from frequency calculations been applied to all reported energy values? It is suggested to clearly label the type of energy used in figures and tables. For example, are the data presented in Fig. 5f and Figure S5 based on Gibbs free energy changes (ΔG) or electronic energy differences (ΔE)? The computational details regarding the type of energy used should also be further clarified in the manuscript.

5. All optimized geometric coordinates from the density functional theory (DFT) calculations should be provided to allow readers and researchers to further validate and reproduce the computational results.

6. In Figure S5, the authors have further supplemented the computational results, indicating that the transition state TS2 and key intermediate III' for the direct attack of amine on difluorocarbene to form the critical C–N bond could not be located in the absence of copper catalysis. To energetically rule out this reaction pathway, it is suggested to perform a C–N bond scan on intermediate II' to validate the feasibility of this route. Moreover, there is an inconsistency in the potential energy profile presented in Figure S5: the energy of intermediate III' is significantly higher than that of the transition state TS2, which is unphysical. This issue needs to be corrected to ensure the reliability and clarity of the proposed mechanism.

Version 2:

Reviewer comments:

Reviewer #3

(Remarks to the Author)

I am satisfied about the revision and responses, and recommend the publication of the revised manuscript.

Response to Reviewer(s)' Comments

Reviewer #1 (Remarks to the Author):

As a versatile synthon, difluorocarbene has received increasing attention recently. While metal difluorocarbene-involved catalytic coupling remains challenging due to the lack of understanding of metal difluorocarbene chemistry. The manuscript by Kang and coworkers reports a copper-catalyzed multi-component reaction via a difluorocarbene pathway. This is a good extension of the pioneering work reported by Zhang (ref 65). The approach uses $\text{BrCF}_2\text{CO}_2\text{K}$ as the difluorocarbene precursor, arylamines as the nucleophiles, and aldehydes as the electrophiles, providing a series of -aminoamide derivatives with moderate to good yields. However, the reaction mechanism is unclear. The authors conducted several experiments to conclude that a copper difluorocarbene species is involved in the reaction. However, the sole control reaction without copper salt cannot support a pathway involving copper difluorocarbene (Fig. 5b). As the difluorocarbene is electrophilic, the direct reaction of arylamine with difluorocarbene, followed by a combination with copper salt, can also generate intermediate I. More evidence to support the copper difluorocarbene-involved pathway would make readers understand the chemistry better. The formation of compound X demonstrates that the attack of arylamine on the difluorocarbene is feasible. The resulting adduct may react with copper salt to generate I. Additionally, it has been shown that copper(I) difluorocarbene is very sensitive to H_2O and decomposes immediately (ref 65). However, the current approach is not sensitive to H_2O , indicating that a copper difluorocarbene might not be involved in the reaction. Overall, this is an interesting reaction. Using difluorocarbene, arylamines, and aldehydes provides an efficient route to -aminoamide derivatives. The manuscript can be published in *Nat. Commun.*, but revisions are needed.

We sincerely appreciate the reviewer's insightful comments and suggestion. Several additional control experiments and density functional theory (DFT) calculations were conducted to further understand the reaction mechanism. These results collectively suggest that the reaction may proceed via a copper-difluorocarbene intermediate (for details, see below).

(1) More mechanistic studies are needed to shed light on the reaction mechanism.

Response: Thank you very much for the insightful suggestion, we performed several additional control experiments and density functional theory (DFT) calculations to shed light on the reaction mechanism:

a) Previous studies (*Nat. Chem.* 2023, 15, 1064–1073; *Org. Lett.* 2016, 18, 4502–4505) have shown that the reaction of copper-difluorocarbene with silyl enol ether do not lead to the gem-difluorocyclopropane, whereas cyclopropanation product is formed in the reaction with free difluorocarbene species. Taking advantage of this characteristic property of copper-difluorocarbene, the competitive experiments of $\text{BrCF}_2\text{CO}_2\text{K}$, p-bromoaniline and silyl enol ether with or without $\text{Cu}(\text{CH}_3\text{CN})_4\text{PF}_6$ were conducted. The formylation product (N-H insertion followed by defluorination) was obtained in 37% yield without detection of

cyclopropanation product under the catalysis of $\text{Cu}(\text{CH}_3\text{CN})_4\text{PF}_6$. However, in the absence of any copper catalyst, difluorocyclopropane formation was unequivocally confirmed by ^{19}F NMR analysis (19% yield), with concurrently observed significantly reduced formylation products ($< 5\%$ yield). The results align with the aforementioned report, confirming the formation of a copper-difluorocarbene intermediate in our reaction system, which likely participates in the multi-component reaction process.

Fig. 1 ^{19}F NMR spectra of difluorocyclopropane and reaction mixture

Fig. 2 HPLC of reaction mixture

b) The reaction of $\text{BrCF}_2\text{CO}_2\text{K}$, p-bromoaniline and benzaldehyde was carried out in the presence of a copper salt and 2,9-diMe-1,10-phen in CH_3CN at 50°C . This condition was reported for the transformations via a copper-difluorocarbene intermediate (*J. Am. Chem. Soc.* 2024, 146, 16902-16911; *Org. Lett.* 2016, 18, 4502-4505). The desired multi-component product was obtained in 60% yield, supporting the potential involvement of a copper-difluorocarbene species in the present reaction.

c) Different Lewis acidic transition metal catalyst including $\text{Sc}(\text{OTf})_3$, $\text{Yb}(\text{OTf})_3$, $\text{Zn}(\text{OTf})_2$, $\text{Ni}(\text{OTf})_2$ and AgSbF_6 were employed as the catalyst in our reaction. The experiments revealed that these alternative metal catalysts either produced only trace quantities ($< 5\%$ yield) of the desired product or failed to promote the reaction altogether. These findings suggest that the copper salt plays a critical and distinctive role beyond conventional Lewis acid catalysis.

entry	[M]	Yield of 4a
1	Sc(OTf) ₃	N.D.
2	Yb(OTf) ₃	N.D.
3	Zn(OTf) ₂	trace
4	AgSbF ₆	trace
5	Ni(OTf) ₂	N.D.

d) Density functional theory (DFT) computations were performed to insight into the reaction mechanism of intermediate IV formation. The free difluorocarbene I complexed with $[\text{Cu}]^+$ to form the copper difluorocarbene II (-20.2 kcal/mol), which is more favored than that directly interactions with ArNH_2 to form a weakly bound van der Waals intermediate II' (-6.2 kcal/mol). The aromatic amine (ArNH_2) then coordinates to the copper difluorocarbene II, providing the intermediate III with a lower free energy about -26.6 kcal/mol. Subsequently, the intermediate III undergoes the C-N formation via transition state TS1 (-26.5 kcal/mol), resulting in the stable intermediate IV. The DFT calculations suggest that the overall process for formation the intermediate IV via copper difluorocarbene is thermodynamically favorable, with an energy release of -28.9 kcal/mol. In the absence of copper catalyst, formation of a weakly bound van der Waals intermediate II' from free difluorocarbene with ArNH_2 was more slowly (-6.2 kcal/mol versus -20.2 kcal/mol). Furthermore, attempts to locate a transition state for C-N bond formation from II' to generate the free ylide intermediate III' were unsuccessful, likely due to a high-energy barrier rendering the process thermodynamically unfavorable. Thus, these results rule out the possibility of intermediate IV forming directly from free difluorocarbene. The DFT calculations highlight the crucial role of copper(I) in facilitating the transformation, wherein coordination with difluorocarbene generates the copper-difluorocarbene intermediate, followed by C-N bond formation to yield IV.

Collectively, the combined evidence from control experiments and density functional theory (DFT) calculations supports the proposed reaction mechanism involving a copper-difluorocarbene intermediate as the plausible pathway. Although copper-difluorocarbene intermediates have been reported to be water-sensitive, the superior nucleophilicity of aniline enables it to outcompete water in reacting with the carbene species. Furthermore, the use of excess carbene precursor and aniline renders the reaction tolerant to trace amounts of water, with minimal impact on the reaction yield under practical conditions. All of the corresponding results were discussed and updated in the revised manuscript and SI.

Fig. 3 DFT computations for the formation of intermediate IV

(2) How about aliphatic amines? The applicability of such amines would improve the synthetic utility of the approach.

Response: Thank you very much for the suggestion. Aliphatic amines such as benzylamine (BnNH₂) and 2,2,2-trifluoroethylamine (CF₃CH₂NH₂) were employed to react with BrCF₂CO₂K and imine under standard conditions. The reactions proceed well to produce the desired products in 36% and 32% yield respectively, albeit with formation of the products that derived from aniline. The corresponding results were added to the revised manuscript (see Fig. 2, 4x and 4y in manuscript).

(3) The sentence presented in the abstract needs to be revised.

“However, its high reactivity and the uncontrollable reaction pathways it triggers limit its widespread application in synthetic chemistry”

Page 2, line 40, “fellow” should be “followed”

Response: We thank the reviewer for pointing out this, the corresponding revision has been updated in the manuscript.

Reviewer #2 (Remarks to the Author):

This manuscript describes a copper-catalyzed synthesis of alpha-aminated amides by connecting arylamines and aromatic or alkyl aldehydes with a carbonyl moiety (CO) derived from a copper difluorocarbene complex. Although the key intermediate of this transformation, a copper difluorocarbene complex, generated from commercially available $\text{BrCF}_2\text{CO}_2\text{K}$ and a copper(I) salt, has already been reported, the authors demonstrated a new way to its application. Furthermore, the reaction proceeds with a broad substrate scope and high functional group tolerance. Because interesting and valuable results are included in this manuscript, its publication in NCOMMS is recommended. Before its acceptance, the authors may consult the following:

We greatly appreciate the reviewer's comments.

(1) Pages 2-3, lines 64-65: The method for the generation of a copper difluorocarbene complex from $\text{BrCF}_2\text{CO}_2\text{K}$ and a copper(I) salt was initially reported in Ref. 61, which Zhang followed in Refs. 65-66. These facts should be clearly described in the manuscript.

Response: Thank you very much for the insightful suggestion, and we have changed the relevant expressions in the revised manuscript.

(2) Page 3, lines 66-68: In the reactions reported in Refs. 62-64, $\text{BrCF}_2\text{CO}_2\text{K}$ and a copper(I) salt are used. However, copper difluorocarbene complexes are not involved, but $\text{CF}_2\text{CO}_2\text{R}$ radical.

Response: We appreciate the reviewer's suggestion. Considering the overall coherence of the article and to maintain a sharper focus on copper difluorocarbene chemistry, we have removed this sentence along with the corresponding references.

(3) Page 3, lines 81-83: The concept: copper difluorocarbene complexes are electrophilic to be attacked by nucleophiles to generate anionic intermediates, which in turn react with electrophiles, is reported in Ref. 61 as a method for ring construction. This fact should be shown in the manuscript by quotation.

Response: We thank the reviewer for pointing out this, the relevant reference has been quoted here.

(4) Page 5, line 124: "4a" should be typed boldly.

Response: We thank the reviewer for pointing out this, the corresponding revision has been updated in the manuscript.

(5) Page 8, line 158: For "cyclohexanol," read "cyclohexanecarbaldehyde."

Response: We thank the reviewer for pointing out this, the corresponding revision has been updated in the manuscript.

(6) Page 8, line 158: For "morphological," read "tetrahydropyran-4-carbaldehyde."

Response: We thank the reviewer for pointing out this, the corresponding revision has been updated in the manuscript.

(7) Page 8, line 158: For “aldehyde esters,” read “glyoxylates” or “glyoxylic esters.”

Response: We thank the reviewer for pointing out this, the corresponding revision has been updated in the manuscript.

(8) Page 9, line 186: Ref. 61 should be quoted before Refs. 65-66 for the method of copper difluorocarbene generation.

Response: We thank the reviewer for pointing out this, the relevant reference has been cited.

(9) Page 9, line 189; Page 10, Fig. 5c: How about alkylamines (RNH₂)? Copper difluorocarbene complexes seem to react with alkylamines to afford isonitriles. If not, alkylamines could be added in Fig. 2 to broaden the substrate scope.

Response: Thank you very much for the insightful suggestion. Aliphatic amines such as benzylamine (BnNH₂) and 2,2,2-trifluoroethylamine (CF₃CH₂NH₂) were employed to react with BrCF₂CO₂K and imine under standard conditions. The reactions proceed well to produce the desired products in 36% and 32% yield respectively, albeit with formation of the products that derived from aniline. The corresponding results were added to the revised manuscript (see Fig. 2, 4x and 4y in manuscript).

Response to Reviewer(s)' Comments

Reviewer #1 (Remarks to the Author):

The authors have effectively addressed the concerns raised by the reviewers. However, the discussion of copper difluorocarbene chemistry in the introduction requires revision. Copper difluorocarbene species was initially proposed by Burton (JACS 1986, 832; and Ref 18). This complex was also suggested by Ichikawa (Ref 61), but there is no direct evidence to confirm the involvement of a $[\text{Cu}(\text{I})]=\text{CF}_2$ species in the reaction. It would be more appropriate to integrate the description of Ichikawa's work into the sentence: "Although $[\text{Cu}]=\text{CF}_2$ complexes were proposed over a decade ago, their development has significantly lagged behind that of $[\text{Pd}]=\text{CF}_2$ complexes."

Response: We thank the reviewer for highlighting this point and the corresponding revisions have been incorporated into the manuscript.

Reviewer #2 (Remarks to the Author):

This manuscript describes a copper-catalyzed synthesis of α -aminated amides by connecting arylamines and aromatic or alkyl aldehydes with a carbonyl moiety (CO) derived from a copper difluorocarbene complex. Because the manuscript was adequately revised, its publication in NCOMMS is recommended. Before its acceptance, the authors may consult the following:

We greatly appreciate the reviewer's comments.

bromodifluoroacetic anion

p-tert butylaniline li

(1) Pages 3, line 7: Quotation "62-64" should be "61-64."

Response: Thank you for pointing out this, and we have made the corresponding revision in the manuscript.

(2) Page 4, Fig 1d: The imine should be the corresponding iminium above the arrow.

Response: We thank the reviewer for pointing out this, the corresponding revision has been updated in the manuscript.

(3) Page 5, line 5 from the bottom: For "underwent well with to react," read "underwent well to react."

Response: We thank the reviewer for pointing out this, the corresponding revision has been updated in the manuscript.

(4) Page 9, line 1: For "p-methylphenyl group (PMP)" read "p-methoxyphenyl group (PMP)."

Response: We thank the reviewer for pointing out this, the corresponding revision has been updated in the manuscript.

(5) Page 9, line 9 from the bottom: For “p-tert butylaniline 1i” read “4-tert-butylaniline 1i.”

Response: Thanks for the reviewer pointing out this, the corresponding revision has been updated in the manuscript.

(6) Page 12, line 13: For “bromodifluoroacetic anion” read “bromodifluoroacetate ion.”

Response: We thank the reviewer for pointing out this, the corresponding revision has been updated in the manuscript.

(7) Page 12, line 13: For “and fellow by” read “followed by.”

Response: We thank the reviewer for pointing out this, the corresponding revision has been updated in the manuscript.

(8) Page 12, lines 14-16: S_NAr substitution with Csp³-F bond should be explained more clearly or by adding a scheme in Fig. 6.

Response: We greatly appreciate the reviewer’s suggesting. Regarding the reaction mechanism, we have now included a more detailed discussion in the revised manuscript.

Reviewer #3 (Remarks to the Author):

In the manuscript, the authors contributed a copper-catalyzed multi-component reaction that efficiently constructs a series of α -aminoamide derivatives using difluorocarbene as a carbonyl source. The authors have preliminarily explored the reaction mechanism through experiments and DFT calculations, proposing that the reaction proceeds via a copper(I) difluorocarbene intermediate. Following the first review, the authors supplemented their work with additional control experiments and DFT calculations to further support the proposed mechanism. The computational part provides some theoretical support for understanding the reaction pathway, particularly for the formation of the copper(I) difluorocarbene intermediate and the subsequent nucleophilic attack by the amine. The manuscript is recommended for publication in Nat. Commun. after further improvements addressing several issues related to theoretical calculations.

We greatly appreciate the reviewer’s comments.

(1) The authors specified in the Supporting Information that the DFT calculations were carried out using the M06-L functional, with geometry optimizations performed with the def2-SVP basis set and single-point energy calculations conducted with the def2-TZVP basis set under the SMD solvation model (acetonitrile). However, for copper-catalyzed reactions, particularly those involving complex electronic structures, it is recommended to compare the results obtained with other functionals—such as ω B97X-D, B3LYP-D3(BJ), or double-hybrid functionals—in order to assess the influence of functional choice on the reliability of the computational results.

Response: Thank you for the reviewer’s suggestion. The formation progress of the copper difluorocarbene [Cu^I]=CF₂ were recalculated with B3LYP-D3(BJ) functional, and all the DFT calculations were performed at the B3LYP-D3/def2-TZVP/SMD(CH₃CN)//B3LYP-D3/def2-

SVP level of theory (details are provided in the supplementary materials). The two different copper configurations for the formation of ammonium ylide intermediate IV were separately considered for application in our calculation process (Nat. Chem., 2023, 15, 1064–1073; Nat. Chem., 2025, 17, 719–726), and the results indicated that the Gibbs free energies of copper difluorocarbene $[\text{Cu}^{\text{I}}]=\text{CF}_2$ II and ammonium ylide intermediate IV for four-coordinated configuration are higher than that for two-coordinated configuration (-11.0 vs -15.3 kcal/mol; -11.2 vs -24.1 kcal/mol). However, we could not obtain the transition state of C-N bond formation for four-coordinated copper configuration. To verify this transition state, potential energy surface scans were performed at 100 different points. The energy fluctuations were found at points 25, 26, 81 and 82, the C-N distances of corresponding the structures are 2.102 Å, 2.123 Å, 3.222 Å and 3.242 Å respectively, which indicate the absence of a C-N bond in the structures at points 81 and 82. For the structures at points 25 and 26, the energy fluctuation are caused by the variation of the dihedral angle between the amine and carbene groups, as the different perspectives were shown below. Thus, the transition state of C-N bond formation via four-coordinated copper configuration could not be located, and two-coordinated copper configuration were used in the DFT calculations. The result has been updated to the revised manuscript and supplementary materials.

Different perspectives of structures at 25 and 26 points :

The computational results with B3LYP-D3(BJ) functional show that free difluorocarbene I preferentially coordinate with $[\text{Cu}^+]^+$ to form the copper difluorocarbene $[\text{Cu}^+]=\text{CF}_2$ II, which is energetically favorable (-15.3 kcal/mol) compared to the direct interaction with ArNH_2 , yielding a weakly bound van der Waals complex II' (4.9 kcal/mol). Subsequent coordination of ArNH_2 to copper-difluorocarbene complex II generates intermediate III (-15.0 kcal/mol), which undergoes C-N bond formation via transition state TS1 (-4.2 kcal/mol) to afford the copper associated ammonium ylide intermediate IV (-24.1 kcal/mol). These results are consistent with our previous DFT calculations outcomes with M06-L functional, and update to the revised manuscript.

(2) In the complete catalytic cycle proposed by the authors (Fig. 6), intermediate IV subsequently reacts with an imine to form V, followed by crucial defluorination and carbonyl migration steps to generate VI and the final product. The current computational study does not cover these critical transformation steps. Particularly, the defluorination step (C-F bond cleavage) and carbonyl migration are vital for a complete understanding of the reaction mechanism. The lack of computational data for these steps leaves the latter half of the mechanism largely speculative. It is recommended that the authors supplement their work with DFT calculations for these key elementary steps, including locating the corresponding transition states and intermediates, and providing the energy profiles.

Response: Thanks for reviewer's suggestion.

According to the control experiments and previous reported, we proposed the reaction mechanism involving sequential ylide interception, defluorination, and carbonyl migration (Acc. Chem. Res. 2013, 46, 2427–2440; Chem. Rec. 2017, 17, 739–753; Acc. Chem. Res. 2023, 56, 592–607). Herein, density functional theory (DFT) calculations were performed to evaluate the energetic feasibility of the proposed reaction process. The nucleophilic ammonium ylide intermediate IV is captured by the activated imine *via* transient state TS2 (1.6 kcal/mol), in which *p*-toluenesulfonic acid and bromodifluoroacetate ion facilitate the proton transfer, leading to the product VII (-54.4 kcal/mol) with releasing the copper(I). The resulting product VII containing a fragment of CF_2 adjacent to nitrogen atom is vulnerable, and the consecutive scission of $\text{Csp}^3\text{-F}$ bond occurs under the assistant of BrCF_2COOK . According to the DFT, the monofluoroimine species VIII (-33.3 kcal/mol) is formed via nucleophilic substitution of

bromodifluoroacetate ion with VII through transition state TS3 (-33.1 kcal/mol), followed by the second Csp³-F bond cleavage to generate intermediate IX (-21.3 kcal/mol). Intramolecular carbonyl migration of intermediate IX via TS5 (-20.6 kcal/mol) with a barrier of 0.7 kcal/mol and subsequent deprotonation eventually render α -aminoamide products. These results have been added to the revised manuscript and supplementary materials.

(3) The first reviewer mentioned that copper(I) difluorocarbene is sensitive to water. The authors explained in their rebuttal that the nucleophilicity of the amine is superior to that of water, and the excess carbene precursor and amine allow the reaction to tolerate trace amounts of water. Although experimental controls (e.g., using 18O-labeled water) ruled out water as the oxygen source for the product, from a computational perspective, comparing the energy barriers for the attack of amine versus water on the copper(I) difluorocarbene could quantitatively assess the possibility and rate differences of these competitive reactions, thereby providing stronger theoretical support for the experimental observations.

Response: We thank the reviewer's suggestion. We calculated the attack of water on the copper(I) difluorocarbene progress (see energy profile below). The free energy of the water-coordinated intermediate III-H₂O is 6.4 kcal/mol high than that of amine-coordinated intermediate III. However, the transition state TS1-H₂O for C-O bond formation could not be located. We conducted an energy scan at 30 points and found energy fluctuations at 15. The C-O and Cu-O distances in the structure at point 15 are 2.814 Å and 2.250 Å respectively, which suggests the absence of C-O bond. Further, we try to obtain the IV-H₂O structure through structure's optimization. We set the 1.475 Å of C-O bond distance, finally the H₂O molecule moves away from the C atom of carbene and coordinates to the copper center. Thus, throughout the scan process and optimization, no TS1-H₂O and IV-H₂O structures were observed. These results suggest that amine is superior to react with copper(I) difluorocarbene than that of water.

(4) When discussing energies, it is recommended to clearly distinguish between Electronic Energy, Gibbs Free Energy, and Enthalpy. The Supporting Information mentions that "refining of single point energies with solvent effect was calculated at the M06L and def2-TZVP level of theory," which usually refers to electronic energies. Have the Gibbs free energy corrections obtained from frequency calculations been applied to all reported energy values? It is suggested to clearly label the type of energy used in figures and tables. For example, are the data presented in Fig. 5f and Figure S5 based on Gibbs free energy changes (ΔG) or electronic energy differences (ΔE)? The computational details regarding the type of energy used should also be further clarified in the manuscript.

Response: We thank the reviewer's suggestion. We show the calculated energies for all the species involved in the reactions in bellow table, and provide the calculated energies to the supplementary materials.

	ZPE	Hc	Gc	E	H	G	Total S	SCF Done
BrCF2COOK	0.025356	0.034912	-0.01224	-3599.81	-3599.8	-3599.85	99.237	-3600.77
TSOH	0.141998	0.152835	0.105912	-894.773	-894.762	-894.809	98.757	-895.701
CH3CN	0.045065	0.049628	0.021032	-132.615	-132.61	-132.639	60.184	-132.825
Cu(CH3CN)3	0.139426	0.153499	0.096155	-2038.08	-2038.07	-2038.13	120.691	-2039
NH2Ar	0.106996	0.115141	0.074504	-2860.59	-2860.58	-2860.62	85.529	-2861.34
PhCNPhBr	0.193144	0.206455	0.151784	-3129.48	-3129.47	-3129.53	115.066	-3130.61
KF	0.001115	0.004652	-0.02093	-699.654	-699.651	-699.676	53.846	-699.908
BrCF2COOKF	0.025828	0.037709	-0.01713	-3699.65	-3699.64	-3699.7	115.42	-3700.82
BrCF2COO-	0.023491	0.030374	-0.0081	-2999.94	-2999.93	-2999.97	80.968	-3000.88
BrCF2COOH	0.036982	0.04473	0.00411	-3000.46	-3000.46	-3000.5	85.493	-3001.34
TSO-Cu	0.17856	0.196625	0.126264	-2667.21	-2667.19	-2667.26	148.088	-2668.59
I	0.007157	0.011083	-0.01688	-237.506	-237.502	-237.53	58.859	-237.805
II	0.056971	0.067083	0.018456	-2010.31	-2010.3	-2010.35	102.346	-2011.16
III	0.166578	0.185128	0.11554	-4870.95	-4870.93	-4871	146.46	-4872.52
TS1	0.165193	0.183237	0.11628	-4870.94	-4870.92	-4870.99	140.923	-4872.5
IV	0.168626	0.186238	0.117541	-4870.95	-4870.94	-4871	144.585	-4872.54
V	0.508966	0.552063	0.424626	-8895.34	-8895.29	-8895.42	268.212	-8898.9
TS2	0.50833	0.550777	0.426152	-8895.3	-8895.26	-8895.38	262.296	-8898.86
VI	0.33077	0.354635	0.271333	-6228.05	-6228.02	-6228.1	175.324	-6230.3
VII	0.317753	0.34166	0.259667	-6227.71	-6227.69	-6227.77	172.57	-6229.88
TS3	0.342546	0.376717	0.268491	-9827.53	-9827.5	-9827.61	227.78	-9830.63
VIII	0.341778	0.372241	0.274004	-9127.82	-9127.79	-9127.89	206.757	-9130.71
TS4	0.367315	0.407678	0.284993	-12727.7	-12727.6	-12727.7	258.213	-12731.5
IX	0.339803	0.368755	0.273342	-9027.81	-9027.78	-9027.87	200.814	-9030.63
TS5	0.339331	0.367793	0.273701	-9027.8	-9027.78	-9027.87	198.033	-9030.63
X	0.340176	0.369264	0.274004	-9027.81	-9027.78	-9027.88	200.493	-9030.64
4	0.327628	0.356594	0.260464	-9027.48	-9027.45	-9027.55	202.324	-9030.23

(5) All optimized geometric coordinates from the density functional theory (DFT) calculations should be provided to allow readers and researchers to further validate and reproduce the computational results.

Response: We thank the reviewer's suggesting, the corresponding revision has been updated in the supplementary materials.

(6) In Figure S5, the authors have further supplemented the computational results, indicating that the transition state TS2 and key intermediate III' for the direct attack of amine on difluorocarbene to form the critical C–N bond could not be located in the absence of copper catalysis. To energetically rule out this reaction pathway, it is suggested to perform a C–N bond scan on intermediate II' to validate the feasibility of this route. Moreover, there is an inconsistency in the potential energy profile presented in Figure S5: the energy of intermediate III' is significantly higher than that of the transition state TS2, which is unphysical. This issue needs to be corrected to ensure the reliability and clarity of the proposed mechanism.

Response: We thank the reviewer's comment. We have performed the C–N bond scan on intermediate III' at 62 different points, and found no energy fluctuations (see below). The potential energy profile in Figure S6 has been updated in the supplementary materials.